# The Role of Mind Wandering During Incubation in Divergent and Convergent Creative Thinking

**DOI:** 10.3390/brainsci15060595

**Published:** 2025-06-01

**Authors:** Qiuyu Du, Rebecca Gordon, Andrew Tolmie

**Affiliations:** Department of Psychology and Human Development, Institute of Education, University College London, 20 Bedford Way, London WC1H 0AL, UK; qiuyu.du.17@ucl.ac.uk (Q.D.); rebecca.gordon@ucl.ac.uk (R.G.)

**Keywords:** mind wandering, incubation, divergent thinking, convergent thinking, creativity

## Abstract

Background/Objectives. While mind wandering has often been linked to negative outcomes, some research suggests it has potential benefits for creativity, particularly through incubation. However, two critical gaps remain: limited understanding of mind wandering’s effects on creative performance and lack of comparative research examining its impact on both divergent and convergent thinking. The study addressed these gaps by comparing the effects of two types of mind wandering (i.e., with and without awareness) on both types of creative thinking, using repeated and novel problems post-incubation to isolate effects. Methods. Eighty-five participants completed divergent (Unusual Uses Task, UUT) and convergent (Compound Remote Associate Task, CRA) thinking tasks, interspersed with a 0-back incubation task. Thought probes measured mind wandering frequency and awareness. Performance was assessed for fluency and originality (UUT) and accuracy (CRA), with problems categorised by difficulty. Results. Results revealed no significant effects of mind wandering on divergent thinking, though incubation improved fluency, particularly for repeated items. For convergent thinking, mind wandering with awareness enhanced performance on low-difficulty repeated items, while mind wandering without awareness hindered novel moderate-difficulty items. Divergent and convergent performance showed no correlation, suggesting distinct cognitive demands. Conclusions. The findings provide evidence that mind wandering’s impact on creativity is limited and context-dependent, with conscious reflection during incubation more beneficial than uncontrolled drifting. Differences in task demands and difficulty levels further modulate these effects. Future research should explore naturalistic settings and use of incubation tasks that do not compete for cognitive resources with the core task to better understand incubation and mind wandering’s roles in creativity.

## 1. Introduction

Mind wandering refers to a shift in the focus of thought away from an ongoing task or events in the external environment to self-generated thoughts or feelings [1]. There has been a substantial body of research on the adverse effects of mind wandering—for example, its negative impact on reading comprehension, where frequent mind wandering leads to reduced understanding [2]. However, there is a gap in the literature regarding its potentially positive effects, particularly its role in facilitating creativity via its connection to the effects of incubation, where diffused attention [3] and reduced latent inhibition (i.e., filtering out of irrelevant information) [4] might allow novel connections to be made between existing information. A further gap is that even where existing studies have tested the connection between mind wandering and creativity, these have predominantly focused on divergent thinking (i.e., generating diverse responses to open-ended questions) with considerable variability in outcomes. The other acknowledged strand of creative cognition, convergent thinking, which focuses on identifying a single correct solution to well-defined problems [5], has largely been neglected. To address these gaps and provide a more holistic view of the relationship between creativity and mind wandering during incubation, the present study examined the effects of mind wandering on both divergent and convergent thinking tasks, taking into account the potentially different effects of mind wandering with and without conscious awareness that might account for previous inconsistencies in research outcomes.

### 1.1. Mind Wandering and Divergent Thinking

The incubation paradigm has been frequently used to explore the role of unconscious processing in creative problem-solving. Baird et al. [6], focusing specifically on mind wandering and divergent thinking, used the interpolated activity method [7], in which participants were given a creative problem (the Unusual Uses Task—UUT) [8] to solve within a specific time period and then provided with an incubation period, following which they returned to the original task [6]. They tested four incubation task conditions: demanding (the 1-back task, where participants had to say whether a presented word matched the word shown on the preceding trial); undemanding (the 0-back task, where the match was to a known target); rest (sitting quietly); and no incubation period. The extent of mind wandering was measured retrospectively using the daydreaming frequency subscale of the Imaginal Process Inventory [9]. The results revealed that improved performance was only observed in the undemanding condition, consistent with the idea that tasks with low cognitive demand promote a beneficial role for mind wandering by allowing space for attention to drift toward musing on the core divergent task material.

In contrast, Smeekens and Kane [10] utilised thought probes (i.e., interruptions during a task where participants are asked to report their current mental state) to measure mind wandering during incubation and tested the impact of varying task demands as in Baird et al. [6] but by using four versions of the N-back task: a 0-back task, a 1-back task, a 2-back and 3-back task, and also reading a written text to occupy attention in another way. Despite variations in both the demand of the incubation tasks and the rate of mind wandering, correlations between mind wandering during incubation and performance on divergent thinking (the Alternative Uses Task—AUT) were consistently null across all conditions. Recent studies by Steindorf et al. [11] and Murray et al. [12] employed a similar incubation condition (the 0-back task) to Baird et al. [6] and again found no advantage for post-incubation divergent thinking performance using the UUT and the AUT. These findings challenge the view that incubation periods, particularly those involving low-demand tasks, facilitate improvements in divergent thinking performance by providing space for mind wandering.

The studies discussed thus far examined mind wandering as a spontaneous and uncontrolled event. However, there is value in comparing the effects of both deliberate and spontaneous mind wandering to understand whether this factor might influence relationships with creative thinking [13]. For example, Seli et al. [14] observed different correlations between these types of mind wandering and mindfulness. Using the aspects of the five-facet mindfulness questionnaire, it was found that non-reactivity to inner experiences had a negative association with spontaneous mind wandering and a positive association with deliberate mind wandering. This suggests those who are able to maintain a focus during inward reflection can make deliberate use of mind wandering and avoid spontaneous drifting, potentially making mind wandering a beneficial tool rather than a deleterious distraction.

Relatedly, a study by Teng and Lien [15] investigated the influence of different types of mind wandering within three contrasting incubation conditions: a 0-back task (mind wandering-prone condition); focused-breathing practice (mindfulness-induced condition); and a control condition where there was no incubation period. Creative performance was measured using the divergent thinking task, the UUT [8]. The findings revealed that participants in the mindfulness-induced condition outperformed those in the mind wandering-prone condition in post-incubation UUT performance, with no incubation effect being observed in the latter. The mindfulness-induced group reported fewer but more diverse mind wandering experiences, including intentional vs. unintentional and aware vs. unaware mind wandering. This heightened awareness and apparently more selective diversity in mind wandering experiences, especially their more deliberate use, may allow for a wider range of ideas to emerge, contributing to creative thinking.

A study by Agnoli et al. [16] supports the argument that intentional mind wandering is important for facilitating creative performance (in this case, measured by the AUT). Mind wandering was assessed retrospectively by two trait-related instruments, the mind wandering-deliberate and mind wandering-spontaneous questionnaires created by Carriere et al. [17]. No association was observed between creative performance and spontaneous mind wandering, but there was a significant association with deliberate mind wandering. However, measurement at the trait level limits the ability to confirm the consistency of mind wandering effects across different contexts since it depends heavily on retrospective memory. Integrating thought probes into an incubation task like the 0-back task might offer a clearer measure of ongoing mind wandering in creative contexts.

### 1.2. Mind Wandering and Convergent Thinking

Few studies directly address the interplay between mind wandering, incubation and convergent thinking. Tan et al. [18] used a number reduction task (NRT) to assess participants’ convergent thinking. Participants viewed digit sequences (e.g., 1, 4, 9) and applied one of two rules: if two successive digits were identical (e.g., 4, 4), they responded with that digit; if the digits differed (e.g., 1, 9), they responded with the unused digit (here, 4). This process was repeated seven times per trial, with a hidden rule being that the final response always matched the second response. Participants’ ability to deduce this rule served as the measure of convergent thinking. There were 90 trials of the NRT before an incubation period, during which mind wandering was measured via thought probe. This was followed by another 300 trials of the NRT. The results showed that those who identified the hidden rule post-incubation reported more mind wandering experiences during the incubation period compared to those who did not. This suggests that mind wandering plays a role in convergent thinking by helping promote insight into task rules.

Leszczynski et al. [19] provided further evidence of the positive effect of mind wandering on this form of creativity using the Compound Remote Associate Task (CRA) [20]. Participants in this task were presented with sets of three words and were required to find a fourth word that could combine with each of the three given words to form a compound word or phrase (e.g., ‘ice’ for ‘cream’, ‘skate’, and ‘water’). After completing a first stage of the CRA task, participants performed the Sustained Attention to Response Task (SART) during the incubation phase. This task requires participants to respond to frequent non-targets while withholding responses to infrequent targets. During this task, thought probes were used to measure rates of mind wandering. After the SART, the participants were again asked to solve CRA problems. The results showed a significant positive correlation between the rates of mind wandering and post-incubation performance and no correlation between mind wandering and pre-incubation performance. This highlights the importance of an incubation break that permits mind wandering for subsequent processing of convergent task material, and since the post-incubation problems were novel, again suggests insight into task structure may have been the key to this.

In contrast, however, Smeekens [21] failed to find a beneficial role for mind wandering in two convergent tasks: the coins problem (i.e., participants need to adjust the position of two coins to make sure each of eight coins touches three others) and the pigpen task (i.e., participants need to build two additional square fences to ensure that each of nine pigs is behind a separate fence). Mind wandering was probed during an incubation period where participants performed a demanding n-back (2-back) task. Results showed no correlation between mind wandering frequencies and post-incubation solution rates, despite the problems being repeated. This suggests that problem type might influence incubation effects, and mind wandering may not be universally beneficial across all convergent task types. Although it requires further investigation, it is possible that tasks involving semantic material (e.g., hidden rules in the NRT or linguistic processing in the CRA) benefit from mind wandering and therefore provide a more productive context for research in this area, whereas performance on spatial tasks (e.g., the coins and pigpen tasks) does not. Note that the tasks used in past research on mind wandering and divergent thinking have characteristically been semantic in nature, so the semantic vs. spatial distinction has not been apparent in that context.

### 1.3. The Relationship Between Divergent and Convergent Thinking

The mixed results on the role of mind wandering in divergent and convergent thinking raise a further question: are there two distinct processes that mind wandering supports in different ways during incubation? Divergent thinking involves exploring multiple potential ideas, which might be facilitated by allowing a range of mind wandering experiences, as in Teng and Lien [15], while convergent thinking involves narrowing down possibilities to find a single correct solution, which may be supported by honing in on key insights into task rules and structure. Despite both being regarded as forms of creative problem solving, the relationship between divergent and convergent thinking remains unclear, as no studies have directly examined possible links between them. However, Duan et al. [22] found that stress affected divergent and convergent tasks in differing ways. While stress was associated with reduced flexibility in divergent thinking tasks, it resulted in lower accuracy and faster reaction time in convergent thinking tasks. This suggests that the two types of thinking are separable, if not separate, processes.

Similarly, de Vink et al. [23] used two types of mathematical tasks: single solution tasks (SST), which require one solution, and multiple solution tasks (MST), requiring multiple solutions, to examine the relationship between divergent and convergent thinking on mathematics problems. In the SST, participants with low divergent thinking scores but high convergent thinking scores showed the highest performance. However, the highest performance in MST was seen in participants with high levels of both divergent and convergent thinking. These findings suggest that the relationship between divergent and convergent thinking might depend on task conditions. Cropley’s [24] summation model supports this idea, emphasising the compensatory relationship between them. That is, strengths in one type of thinking may moderate the effects of the other, which explains the results of de Vink et al.’s study [23]: strong convergent thinking is needed for SSTs, and strong divergent capabilities may be a distraction, while good performance in MSTs requires strong divergent thinking to generate possible solutions and strong convergent thinking to narrow these down to those that are most appropriate. Together, these findings suggest that the relationship between divergent and convergent thinking is not fixed but dynamic, based on the specific context, while their apparent separability suggests that they might be supported by mind wandering in different ways.

### 1.4. The Current Study

The present study investigated the role of mind wandering during incubation in divergent and convergent thinking tasks, using a repeated-measures design. While past research has explored mind wandering’s role in creativity, two key gaps motivated this work: limited evidence on the positive effects of mind wandering on creative performance and a lack of comparative studies examining its impact on both divergent and convergent thinking. This study aimed to address these gaps by testing how mind wandering during incubation influences these cognitive processes. All participants completed both divergent (the Unusual Uses Task) and convergent thinking (Compound Remote Associate Task) tasks, with the order counterbalanced. These tasks were selected to be comparable in terms of their semantic character, approximate duration and item granularity, while contrasting in terms of their requirement for wide-ranging reflection vs. insight into task structure and concomitant strategy. The convergent thinking task also differentiated between items at three levels of difficulty, based on the pass rate observed in past research using the CRA by Wu and Chen [25], making it possible to identify whether there was a specific difficulty threshold at which incubation effects become apparent. Randall et al. [26] and Xu and Metcalfe [27] argue that low-difficulty problems do not benefit from incubation or mind wandering since performance does not require additional support, while high-difficulty problems are too complex for incubation or mind wandering to provide substantial help. It was not possible to differentiate the difficulty level for the divergent thinking task due to the lack of past comparison of pass rates on different items.

Each task consisted of three phases: an initial performance phase; an incubation phase during which participants completed the 0-back task for approximately 15 min, when they were probed intermittently for the occurrence of mind wandering; and finally, a second performance phase, with both repeated and novel items. Employing repeated and novel items facilitated the separation of the specific effects potentially induced by mind wandering during the incubation period from effects of insights into task structure and strategy. Improved performance on repeated items only would suggest incubation effects with respect to specific problems, while improvement on novel items compared to the baseline might suggest wider effects. The 15 min incubation period was based on reviews suggesting that this duration is sufficient to capture incubation effects without diminishing performance due to overly long breaks [28,29].

It was hypothesised that mind wandering would have differing effects on performance in divergent and convergent thinking tasks. On the basis of the research discussed above, the expectation for divergent thinking was that mind wandering with awareness would support the generation of creative ideas, but mind wandering without awareness would not [15,16]. For convergent thinking, it was hypothesised that mind wandering with awareness would facilitate post-incubation performance on the CRA task, but that mind wandering without awareness might have negative effects due to distraction, extrapolating from de Vink et al. [23] on the negative impact of divergent thinking during convergent tasks. It was also expected that there would be a relationship between difficulty levels and incubation effects, with the benefits of mind wandering during incubation being highest at the moderate level of difficulty, forming an inverted U-shaped effect [26,27]. Finally, given that different cognitive processes are involved in divergent and convergent thinking, it was hypothesised that individuals’ abilities in one type of thinking would not strongly predict their abilities in the other, resulting in at most a modest correlation between them.

## 2. Materials and Methods

### 2.1. Participants

A power analysis revealed that 84 participants were required to detect medium-sized effects on both repeated *t*-tests and bivariate correlations with a power of 0.8. Eighty-five participants (mean age = 33.68 years; SD = 12.69, 40 female) were recruited from a Chinese university and a food processing enterprise located in the east of China. Recruitment was conducted via advertisements on social media and staff networks. Participants’ eligibility was determined based on a lack of any history of cognitive impairments or neurological illness, ensuring that those recruited possessed the necessary cognitive capacity and health status for the successful completion of the study’s tasks. A broad sample was targeted to enhance the generalisability of the study. All participants were presented with an information sheet, and informed consent was obtained before commencing with testing. The study was approved by the authors’ institutional Research Ethics Committee.

### 2.2. Materials

#### 2.2.1. Divergent Thinking

The Unusual Uses Task (UUT) was used to measure divergent thinking. In the original UUT task, participants were presented with a set of everyday objects (e.g., bricks, newspapers, spoons) and were asked to generate as many potential non-standard uses as possible for each item (e.g., using a brick as a paperweight) within a specified time limit [8]. In the present instance, they were given 6 min to complete a baseline task, consisting of two UUT items: a brick and a plastic bottle. After a period of incubation (described below), the task was repeated with the baseline items, plus two new UUT items (newspapers and a pair of chopsticks), with 12 min to complete the task.

An objective scoring method was used due to its reliability and convergent and predictive validity [30]. Responses given by less than 20% of the participants were classified as original and awarded one point. Other responses were scored 0 for originality. Given the potential problems of scoring less frequent responses as original without a quality check, fluency, the total number of responses produced regardless of their originality, was also scored to examine incubation effects on the quantity of ideas generated. Improvement pre- to post-incubation was calculated as the raw difference between scores at the two time points, with separate measures being derived for originality and fluency. Pre-incubation scores were used as the baseline for both repeated and novel items post-incubation.

#### 2.2.2. Convergent Thinking

The Compound Remote Associate Task (CRA) was used to measure convergent thinking. In the baseline task, participants were presented with sequences of three words and, in each case, asked to make a remote association with a target fourth word to form meaningful word phrases. Eight CRA problems were chosen from Wu and Chen’s [25] Chinese CRA problem set. These were categorised into three levels of difficulty as determined by pass rate in that research: 0.5–0.8 (low difficulty, comprising 2 items), 0.3–0.5 (moderate difficulty, comprising 3 items), and 0.1–0.3 (high difficulty, comprising 3 items). Participants had six minutes to complete the set of eight problems. The post-incubation task involved the same CRA items presented in the baseline task plus eight new items sharing the same variation in level of difficulty, with 12 min to complete the task. Each correct answer (i.e., where the target answer was given) received one point. Improvement on both repeated and novel items post-incubation was again calculated as the raw difference from baseline performance. Separate values were obtained for overall improvement and for improvement within each difficulty level.

#### 2.2.3. Incubation

The 0-back task was used for the incubation task. Participants were presented with a target word and, over a succession of subsequent trials, asked to decide whether a further word matched this target. The task consisted of 15 blocks, and each block contained 16 trials, resulting in a total of 240 trials. Target stimuli (i.e., matches with the pre-specified word) were pseudo-randomly inserted in a block. The ratio of non-target to target items was 8 (1 target per 8 trials). The mean task duration was 15 min, with variation due to participants’ response time and pace through the trials. Two measures were used to assess consistency of performance across the incubation periods for the divergent and convergent tasks: accuracy and reaction time. Accuracy was calculated as the proportion of correct responses out of all trials (i.e., hits on target trials and correct rejections on non-target trials), with a maximum score of 1. Reaction time was the mean RT across all trials.

#### 2.2.4. Thought Probes

Participants were given thought probes during the incubation task to determine the frequency and nature of their mind wandering. The probes were presented at pseudo-random intervals, so participants were not able to anticipate when they were actually going to happen. On each occasion, they were given five options to select from to identify the nature of the mind wandering or other activity: (1) attention to the task at hand, (2) external distraction, (3) task-related reasoning, (4) mind wandering without awareness, and (5) mind wandering with awareness. Options 4 and 5 were used to record the frequency of mind wandering of each type.

### 2.3. Procedure

For the purposes of counterbalancing, participants were split into two groups: (1) the divergent thinking task followed by the convergent thinking task and (2) the convergent thinking task followed by the divergent thinking task.

For the divergent thinking task, participants were first asked to complete two UUT problems, followed by the incubation task (including thought probes). Participants then completed the UUT problems again, with two additional novel problems interspersed between the repeated items. They could allocate time freely between repeated and novel problems.

For the convergent thinking task, participants were first asked to complete eight CRA problems, followed by the incubation task (with probes). They then completed the CRA problems again, interspersed with an additional eight novel items.

Both thinking tasks had a mean administration time of 33 min. Participants proceeded directly from one task to the other without a delay between the two. Instructions and task stimuli were available in both digital (i.e., on-screen) and paper formats. Participants were presented with this choice prior to the commencement of the task, allowing them to choose their preferred response method to ensure optimal performance. This experiment was conducted face-to-face in a quiet area away from distraction.

### 2.4. Analysis Plan

Preliminary analyses were conducted to assess whether there were any significant differences in performance between (a) those who responded digitally and on paper; (b) student vs. non-student participants; and (c) the contrasting divergent and convergent task orders.

To compare pre- and post-incubation performance on the UUT task, improvement from baseline to post-incubation was analysed for repeated and novel items, with repeated *t*-tests comparing these conditions to examine incubation effects; use of ANOVA to examine pre- to post-incubation improvement by item type was not possible because of the shared pre-incubation baseline scores. For the CRA task, the improvement baseline to post-incubation was analysed by item difficulty, with two-way repeated measures ANOVA and follow-up *t*-tests being used to compare the effects of the different exposure conditions and levels of difficulty. Correlation analyses explored associations between mind wandering and CRA performances, including improvement scores, across both exposure conditions. Correlation analyses and *t*-tests were conducted to compare creative performance (UUT and CRA), the 0-back task, and the incidence of mind wandering between the convergent and divergent sections. Finally, correlation analyses were employed to test associations between mind wandering and improvement on the performance indicators across the different exposure conditions in both tasks.

## 3. Results

### 3.1. Preliminary Analyses

It was necessary to assess the impact on performance of occupation (students vs. others) and response mode (digital vs. paper-pencil) independently since response mode distribution was uneven across participant groups, with the vast majority of students responding digitally but a two-thirds/one-third split in favour of digital responses in the non-student group. The only significant difference that was identified was that between response modes in baseline convergent thinking performance, with the paper-pencil group outperforming the digital group, Mann–Whitney U = 1045.5, N_digital_ = 61, N_paper-pencil_ = 24, *p* = 0.002. However, there was no significant difference between groups in improvement from baseline to post-incubation, t(83) = 1.38, *p* = 0.17, indicating that the baseline difference had no impact on any incubation effects. The paper-pencil format may have initially enhanced focus and reduced distractions, benefiting goal-directed and structured problem solving in the convergent task, but there were no carry-over effects from this on the outcomes of interest. Given this and the lack of other effects on outcomes, both response mode and participant group were discounted from further consideration in subsequent analyses.

To test whether improvement differed based on task order (divergent-first vs. divergent-second), a two-way mixed ANOVA on improvement score was employed, with repeated vs. novel exposure as a within-subjects factor and task order as a between-subjects factor. There were no effects of task order on improvement scores for either the divergent or convergent task, and this factor was therefore also discounted from further consideration.

### 3.2. Divergent Thinking Task

There was a significant increase from baseline to post-incubation in fluency scores on the UUT task. This was the case for both repeated exposure items, t(84) = −9.53, *p* < 0.001, d = 1.03, and novel exposure items, t(84) = −4.35, *p* = 0.001, d = 0.47 (see Table 1). The improvement in fluency was significantly greater for repeated exposure items than for novel ones, t(84) = 2.79, *p* = 0.006, d = 0.30. Originality scores also improved significantly between baseline and post-incubation for the repeated, t(84) = −10.16, *p* = 0.001, d = 1.10, and novel items, t(84) = −4.73, *p* = 0.001, d = 0.52. However, the improvement in originality scores did not differ significantly between the two item types, t(84) = 1.59, *p* = 0.12, d = 0.17. Creative performance improved therefore on both repeated and novel exposure items. However, the degree of progress varied across the two conditions, particularly in terms of fluency scores, where the greater improvement on the repeated items was consistent with the presence of item-specific incubation effects. The lack of significant difference in the improvement in originality scores between the two types of items suggests that any such effect resulted in greater quantity rather than quality of ideas, however.

### 3.3. Convergent Thinking Task

For CRA performance, there was a significant increase from baseline to post-incubation in scores for the repeated exposure condition, t(84) = 5.59, *p* = 0.001, d = 0.37, and the novel exposure condition, t(84) = 2.58, *p* = 0.01, d = 0.33 (see Table 2). There was no significant difference between the two conditions, t(84) = 0.35, *p* = 0.72, d = 0.04, suggesting change resulted from possible insight rather than item-specific incubation effects. This was further supported by the finding that there was a significant negative correlation between baseline and raw change scores in the novel exposure condition, r(83) = −0.47, *p* < 0.001, suggesting an insight effect for those who performed poorly in the pre-incubation trials.

Analysis of task difficulty in each condition revealed a somewhat more nuanced pattern of effects. A two-way repeated measures ANOVA, where the first factor had two levels (exposure type: repeated vs. novel) and the second factor had three levels (difficulty level: low, moderate, high), was used to look at the effects of exposure type and difficulty level on proportional improvement scores (necessitated by the difference in number of items). To calculate proportional improvement scores, raw change scores were divided by the maximum possible change from baseline to post-incubation (i.e., 2 for low difficulty, 3 for moderate, and 3 for high). This produced a normalised proportional measure, allowing fair comparison across difficulty levels. The main effect of exposure was not significant, F(1,84) = 0.54, *p* = 0.46, partial η^2^ = 0.01, indicating that there was no difference between repeated and novel exposure conditions in terms of proportional improvement. The main effect of difficulty was significant, F(2,166) = 20.17, *p* < 0.001, partial η^2^ = 0.20, indicating that performance varied significantly across difficulty levels. There was a significant interaction between exposure types and difficulty level, F(2,166) = 15.40, *p* < 0.001, partial η^2^ = 0.16.

As the interaction between exposure types and difficulty level was significant, the main effect of exposure at each level of difficulty was examined, reverting to use of the raw scores as the more precise metric. There was a significant gain in performance for the low-difficulty items from baseline to post-incubation for the repeated problems, t(84) = 2.62, *p* = 0.01, d = 0.28, but performance on novel exposure items decreased significantly, t(84) = −2.33, *p* = 0.02, d = 0.25. The difference in change between the two conditions was significant, t(84) = −3.15, *p* = 0.002, d = 0.34, consistent with gains at this level of difficulty being due to item-specific incubation effects.

For moderate difficulty items, significant increases were observed in scores for both repeated exposure problems, t(84) = 4.31, *p* < 0.001, d = 0.47, and novel exposure problems, t(84) = 6.64, *p* < 0.001, d = 0.72. The improvement in the repeated exposure condition was significantly lower than in the novel, t(84) = −3.78, *p* < 0.001, d = 0.41, consistent with the effects of insights into problem structure and strategy at this level of difficulty. These appeared to benefit performance on the novel items to a greater extent, perhaps because these were unconstrained by previous thinking.

For the high-difficulty items, performance on repeated problems improved significantly from baseline to post-incubation, t(84) = 3.94, *p* < 0.001, d = 0.43. However, the difference under the novel exposure condition was not significant, t(84) = 1.38, *p* = 0.17, d = 0.15. The difference in improvement between the two conditions was also not significant due to high levels of individual variability in the extent of change, t(84) = 0.96, *p* = 0.34, d = 0.10. Again, this might be explained by any gains being primarily due to effects of generic insight, rather than the impact of incubation on specific items.

To summarise, participants showed significant overall improvement from baseline to post-incubation in both repeated and novel exposure conditions, but with effects of incubation more evident on low-difficulty items and changes at the moderate and high-difficulty levels more attributable to insight effects. The lack of incubation effects at higher levels of difficulty might be because these problems are too challenging to benefit from reflection on specific solutions and benefit more from a focus on broader strategy. The reason for the decline in the low-difficulty novel items is less clear, but this might be attributable to a greater focus on specific solutions as opposed to strategy, coupled with the novel post-incubation items proving more difficult than the baseline ones.

### 3.4. Comparisons Between the Divergent and Convergent Tasks

There was a high level of consistency between measures taken during the incubation period for the divergent and convergent tasks. For the 0-back task, *t*-tests showed no significant differences between divergent and convergent sections in reaction time, t(84) = 0.90, *p* = 0.37, d = 0.09, or accuracy rate, t(84) = −1.01, *p* = 0.32, d = −0.13. There were also significant positive correlations for reaction time, r(83) = 0.60, *p* < 0.001, and accuracy rate, r(83) = 0.93, *p* < 0.001, between the tasks. Similarly, the incidence of mind wandering during the incubation period showed no variation between the divergent and convergent tasks (see Table 3). There were no significant differences between the two periods in overall mind wandering, t(84) = −1.41, *p* = 0.16, d = −0.15; mind wandering with awareness, t(84) = −1.17, *p* = 0.25, d = −0.13; or mind wandering without awareness, t(84) = −0.99, *p* = 0.32, d = −0.11. Overall mind wandering during the incubation period for both task types was positively correlated, r(83) = 0.47, *p* < 0.001, as was mind wandering with awareness, r(83) = 0.39, *p* < 0.001, and mind wandering without awareness, r(83) = 0.51, *p* < 0.001. These findings suggest that participants’ tendency toward mind wandering was stable across the two tasks. The effectiveness of the 0-back task in allowing mind wandering was confirmed by correlations between mind wandering and reaction time. Participants’ 0-back reaction time during incubation for the divergent task was positively correlated with mind wandering (overall: r = 0.33, *p* = 0.002; with awareness: r = 0.26, *p* = 0.02; without awareness: r = 0.27, *p* = 0.01). Similarly, there were significant positive correlations between reaction time and mind wandering during incubation for the convergent task (overall: r = 0.25, *p* = 0.02; without awareness: r = 0.22, *p* = 0.048), though not for mind wandering with awareness (r = 0.19, *p* = 0.08). In general, then, the more slowly participants responded during the 0-back task, the more space they apparently had for mind wandering. There was no correlation between 0-back accuracy and mind wandering during divergent incubation (overall: r = −01, *p* = 0.95; with awareness: r = 0.03, *p* = 0.78; without awareness: r = −0.07, *p* = 0.55) or convergent incubation (overall: r = −0.14, *p* = 0.22; without awareness: r = 0.13, *p* = 0.23), but there was a significant negative correlation for mind wandering with awareness, r = −0.27, *p* = 0.01, suggesting that closer attention to the incubation task acted to some extent to suppress mind wandering.

In contrast to this general pattern of close similarity between the two incubation tasks, correlational analysis showed no significant association between convergent and divergent performance scores at baseline (fluency: r(83) = 0.05, *p* = 0.63; originality: r(83) = 0.11, *p* = 0.31). For post-incubation performance, there was no significant relationship between convergent and divergent scores on repeated exposure problems (fluency: r(83) = −0.03, *p* = 0.79; originality: r(83) = −0.04, *p* = 0.74) or novel exposure problems (fluency: r(83) = −0.08, *p* = 0.48; originality: r(83) = −0.03, *p* = 0.76). Similarly, for improvement scores, the analysis showed non-significant correlations between convergent and divergent tasks in the repeated exposure condition (fluency: r(83) = −0.08, *p* = 0.44; originality: r(83) = −0.14, *p* = 0.20) and the novel exposure condition (fluency: r(83) = 0.02, *p* = 0.85; originality: r(83) = 0.01, *p* = 0.97). The highly consistent performances on the 0-back task and mind wandering indicate that this lack of relationship between convergent and divergent tasks is a genuine effect rather than the product of random variation in performance.

### 3.5. Mind Wandering and Improvement on the UUT Task

Despite the evidence in favour of incubation effects, there were no statistically significant associations between mind wandering of any specific type and post-incubation improvement in fluency or originality scores for repeated or novel exposure items, contrary to our hypothesis.

### 3.6. Mind Wandering and Improvement on the CRA Task

For repeated exposure items, there were no significant associations between overall improvement scores and mind wandering, either with or without awareness. However, for low-difficulty problems, where there were the clearest signs of incubation effects, there was a significant correlation between overall mind wandering and improvement, r = 0.28, *p* = 0.008, and between mind wandering with awareness and improvement, r = 0.39, *p* < 0.001, in line with our hypothesis. There was no significant correlation between improvement and mind wandering without awareness. There were no significant associations with mind wandering for moderate- and high-difficulty problems.

For novel exposure items, there were no significant correlations between mind wandering and overall improvement scores, and the same was also the case for low- and high-difficulty problems. For moderate-difficulty problems, however, where there was greatest evidence of insight effects, there was a significant negative association between improvement and mind wandering without awareness, r = −0.28, *p* = 0.01, indicating that this interfered with those effects.

To summarise, under repeated exposure conditions, mind wandering, particularly with awareness, was positively associated with improvement in convergent thinking scores for low-difficulty problems but had no effect on moderate- and high-difficulty problems, suggesting that greater difficulty required more focused attention compared to the low-difficulty problems. Under novel exposure conditions, significant improvement was observed only in the moderate-difficulty problems, where mind wandering without awareness was negatively associated with improvement, consistent with the hypothesised effects of distraction.

## 4. Discussion

The purpose of the present study was to examine the effects of mind wandering during incubation on divergent and convergent creative thinking under comparable low-demanding conditions, using the 0-back task. Repeated and novel items were used post-incubation to differentiate effects of incubation regarding problems already seen by participants from more generalised effects of insight into task structure and strategy. We also examined the relationship between performance on divergent and convergent creative thinking tasks beyond any effects of mind wandering and incubation.

Outcomes supported the hypotheses in limited respects. Mind wandering generally (i.e., with or without awareness) had no significant impact on divergent thinking, though incubation did lead to increased fluency despite this, especially on repeated items. Mind wandering with awareness did have a positive effect on convergent task performance for repeated items but only at low-difficulty levels, not at moderate levels as had been anticipated. Mind wandering without awareness had a distracting effect, but only on novel items at moderate-difficulty levels. As with divergent task performance, there were gains beyond item-specific incubation at both moderate- and high-difficulty levels. Underscoring the disparities in the effects of mind wandering, performance on the divergent and convergent tasks was not even modestly correlated, despite the equivalence between the two task sections on all indices related to frequency of mind wandering and performance during the incubation task.

### 4.1. Factors Contributing to Lack of Effects in Divergent Thinking

There is a need to account for the lack of mind-wandering effects on divergent thinking, contradicting Baird et al. [6], despite having differentiated conscious from unconscious mind wandering. A potential explanation is that, although the present study replicated Baird et al.’s use of 0-back tasks during the incubation period, we observed a higher level of mean accuracy on this (0.96 vs. 0.87). Null effects of mind wandering were also found in studies by Steindorf et al. [11] and Murray et al. [12], where reported accuracies were approximately 0.93 and 0.91. It seems plausible that where participants sustain greater attention on the 0-back task, this places greater demands on cognitive resources, limiting the capacity available for mind wandering and creative thinking during incubation periods. The negative correlation between accuracy during the convergent incubation task and mind wandering with awareness is consistent with this.

Extending this argument, the 0-back task involved verbal processing, as did the creative task. The similarity in processing resources may have led to a competition between mind wandering or creative processing and the incubation task itself. Support for this comes from Gilhooly et al. [31], who investigated the impact of verbal and spatial incubation tasks on varying creative tasks, using verbal and spatial versions. They found that verbal incubation tasks enhanced performance on spatial creative tasks more than on verbal ones, while spatial incubation tasks had the reverse effect. These results indicate that incubation tasks and creative tasks are most effective when they utilise different cognitive resources, minimising interference and competition (cf. Baddeley [32]), and allowing greater scope for mind wandering to have an impact on creative performance.

These points notwithstanding, there was no difference between 0-back performance during the incubation period for the divergent and convergent thinking tasks, and no difference in the extent of mind wandering during incubation. Despite this, the null effect of mind wandering was not repeated for convergent performance. Some non-artefactual account of outcomes for the divergent task therefore seems preferable, especially since the incubation period did lead to improved outcomes, particularly for the repeated items. This might be interpreted as being simply the combined effect of practice at the task and familiarity with those items. The pattern of effects on the convergent task suggests not, however, since the benefits of mind wandering during incubation for repeated items were limited to the simplest problems, with reflections on and insights into task structure appearing to be of greater importance overall, as we discuss next.

### 4.2. Effects on Convergent Thinking

The positive effects of mind wandering with awareness at the lowest difficulty level suggest that deliberate relaxation of focus facilitated responses to previously seen problems that were relatively easy, provided this relaxation remained within conscious control. This aligns with findings by Zedelius and Schooler [33], who found that deliberate mind wandering exhibited a positive correlation with creative performance (see also [16,34]). At the moderate level of difficulty, however, mind wandering without awareness had a negative influence on performance for novel items, suggesting a disruptive effect and the importance of maintaining focus, also consistent with results of previous research [16,17,35]. Specifically, this aligns with findings that mind wandering without awareness activates the neural default network, diverting cognitive resources away from evaluation [36,37]. That is, brain regions required to monitor and evaluate creative processing are occupied, leading to poor evaluation of creative ideas and outcomes. At the moderate level of difficulty, any positive effects of the incubation period appeared, therefore, to have more to do with managed reflection, with benefits for both novel and repeated items. This constrained focus appeared to help participants better understand problem structures and develop solutions. This led to better performance on novel items that were not too difficult, especially for those who performed worse to begin with, and provided some fresh insight into such problems where they had seen these before. Diffuse conscious reflection on the content of the latter either did not occur or was not productive. At the high level of difficulty, the combination of problem insight and previous time spent thinking about specific problems was apparently helpful; insight on its own was not, possibly because these problems were too challenging for this to be productive.

This process of managed reflection may also have been what led to improvement post-incubation on the divergent task, with the added bonus of prior familiarity for the repeated items, as with the high-difficulty convergent items. This implies that incubation is primarily beneficial when it involves a degree of focus, as in Teng and Lien’s [15] mindfulness condition. Spontaneous, unintended drifting is disruptive to maintaining focus. The further implication is that while there are benefits of mind wandering during incubation, they are nevertheless only a limited part of what is involved. It is also important to note that because these different types of impact all stemmed from activity during the one incubation period, it seems plausible that participants engaged in different types of reflection at different moments within it. This highlights the methodological benefits of breaking the task down by different levels of difficulty, for the convergent items at least, since this helped to expose the complexity of processing that occurs during incubation.

### 4.3. Differences Between Divergent and Convergent Thinking

The effect of mind wandering may have been more prominent in convergent problem solving than in divergent, due to the basic nature of the creative tasks employed. Indeed, since the experimental structure, the sample, the performance on the 0-back task, and the extent of mind wandering were all comparable across both studies, only the differential nature of divergent and convergent thinking appears sufficient to explain the observed differences. Task-related variations in memory demands appear to be particularly implicated. The generation of a new idea or concept in the creative thinking process often involves reconstruction based on knowledge extracted from long-term memory, accompanied by sustained attention and cognitive control, all of which involve working memory [38]. Working memory capacity (WMC) has been shown to be positively correlated with convergent performance, particularly with tasks requiring complex problem manipulation [38,39]. Lin and Lien [40] employed both divergent and convergent tasks and found that participants’ WMC correlated with convergent task performance but not with divergent task performance, suggesting that two forms of creativity involve different cognitive resources. A meta-analytic study further confirms a positive relationship between working memory and convergent thinking but no specific relationship with divergent thinking [41].

Thus, it is possible that the differences stem from convergent thinking requiring more or different working memory resources compared to divergent thinking. Convergent tasks typically demand focused attention to narrow options to a single answer, relying heavily on working memory for maintenance and manipulation of information following predetermined guidelines or limitations, hence the restriction of benefits of mind wandering with awareness to the simplest repeated items and the disruptive effects of mind wandering without awareness. Divergent activities prioritise creativity, exploration, and the development of a wide variety of options rather than the selection of a single correct solution and may not demand the same amount of concentrated attention or cognitive resources as convergent tasks, making managed reflection on task structure and strategy potentially easier to achieve. These differences in processing may be what led to the lack of correlation between divergent and convergent performance, with some individuals better at maintaining the more concentrated focus required by convergent thinking and others better at the looser reflection that assists divergent thinking.

### 4.4. Limitations

It must be acknowledged that the current study might be limited by potential differences in how mind wandering manifests in laboratory tasks versus real-life settings. Laboratory tasks may suppress natural mind wandering, as participants are more likely to be maintaining maximum engagement on tasks to avoid errors, contrasting with real-life settings where mind wandering occurs more freely. Related to this are findings that mind wandering in natural settings is linked to divergent thinking [6,42]. This discrepancy might contribute to the explanation of why the divergent study failed to replicate the effects reported by Baird et al. [6], though it perhaps also underscores the partial and relatively fragile nature of mind wandering effects during incubation. Moreover, as here, experimental tasks often isolate single aspects of creativity, focusing exclusively on divergent or convergent thinking, whereas real-life problems typically require an interplay of both and potentially other processes beyond these. While we selected the current divergent and convergent tasks for their semantic comparability and established validity, they nevertheless remain single indicators of their respective constructs. There is therefore a need to develop assessments that mirror real-life problems to yield more valid results.

We also acknowledge that the convergent difficulty level analysis was only based on a few items per level, which may reduce reliability. Future work should include more items per difficulty level to establish robust effects. Further, although performance on the 0-back task was controlled, unmeasured individual differences in effort during this incubation task, potentially linked to different group membership within the sample, may have moderated mind wandering effects. Preliminary analyses showed no group differences in creativity, suggesting there was limited bias in this respect. Nevertheless, broader sampling would enhance generalisability, and a larger sample would be better powered to identify some of the more complex effects that might potentially be at work, given the nuances we uncover.

We acknowledge too that while our thought probes effectively captured the awareness dimension of mind wandering, their brief format necessarily limited assessment of other aspects (e.g., thought content). More extensive probing would have risked disrupting the cognitive processes under study. This trade-off between measurement breadth and ecological validity represents an inherent challenge in mind-wandering research that future studies might address through alternative methodologies. Finally, as noted above, there is a need for investigation of incubation effects that avoid an overlap of cognitive resources between incubation and creative tasks [31].

## 5. Conclusions

This study explored the relationship between mind wandering during incubation and creative problem solving, comparing effects on divergent and convergent creative thinking and examining associations between the two. In contrast to much past research, the results highlight the relatively restricted and nuanced nature of mind wandering effects during incubation. There were no observable effects of mind wandering on divergent thinking and only limited positive effects on convergent thinking at low levels of problem difficulty. However, incubation had effects beyond mind wandering, especially where participants were able to maintain focused reflection and avoid distraction from unconscious drifting of attention. These effects were similar for both divergent and convergent tasks and comprised insights into problem structure and solution strategy, aided by familiarity where problems had been seen before. Despite this similarity in outcomes, performance on divergent and convergent tasks was unrelated, perhaps because of variations in the demands they place on core cognitive systems and individual differences in managing these. Future research needs to examine more carefully the relationship between divergent and convergent thinking and the effects of incubation, including conscious and unconscious mind wandering, at different levels of problem difficulty using ecologically valid tasks.

## Figures and Tables

**Table 1 brainsci-15-00595-t001:** Mean scores (standard deviation) for performance on the UUT.

	Pre-Incubation	Post-Incubation		Improvement	
	Baseline	Repeated	Novel	Repeated	Novel
Fluency	7.71 (2.76)	10.46 (3.89)	9.28 (4.07)	2.75 (2.66)	1.58 (3.34)
Originality	6.55 (2.09)	8.36 (2.55)	7.88 (2.99)	1.81 (1.64)	1.33 (2.59)

**Table 2 brainsci-15-00595-t002:** Mean scores (standard deviation) for performance on the CRA.

Difficulty Level	Baseline Scores	Post-Incubation Scores		Improvement	
		Repeated Exposure	Novel Exposure	Repeated Exposure	Novel Exposure
Overall	1.61 (1.22)	2.13 (1.56)	2.06 (1.49)	0.52 (0.85)	0.45 (1.60)
Low difficulty	0.84 (0.72)	0.93 (0.74)	0.58 (0.70)	0.09 (0.33)	−0.26 (1.03)
Moderate difficulty	0.62 (0.72)	0.86 (0.90)	1.29 (0.92)	0.24 (0.50)	0.67 (0.93)
High difficulty	0.15 (0.36)	0.33 (0.56)	0.25 (0.55)	0.18 (0.41)	0.09 (0.63)

**Table 3 brainsci-15-00595-t003:** Mean frequency (standard deviation) of mind wandering in the divergent and convergent incubation periods.

	Mind Wandering	Mind Wandering with Awareness	Mind Wandering Without Awareness
Divergent section	0.13 (0.19)	0.09 (0.14)	0.04 (0.09)
Convergent section	0.16 (0.23)	0.11 (0.17)	0.05 (0.11)

## Data Availability

The data that support the findings of this study are openly available in the Open Science Framework Repository at https://osf.io/k6p7w/?view_only=60ca7847fb844a89b24b7dd73f65fa1e, accessed on 7 April 2025.

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
