# Peer review of "The Role of Mind Wandering During Incubation in Divergent and Convergent Creative Thinking"

_brainsci, 2025, doi:10.3390/brainsci15060595_

Round 1
Reviewer 1 Report
Comments and Suggestions for Authors
The manuscript explores the impact of mind wandering during an incubation period on divergent and convergent thinking. It involved 85 participants in a repeated-measures design. It incorporates a 0-back incubation task and thought probes to assess mind wandering frequency and awareness. The authors posited that mind wandering with distinct effects on divergent and convergent thinking.
Although the study has some merits, it also exhibits significant flaws:
1. For divergent thinking, no significant effects of mind wandering were observed. It contradicts studies like Baird et al. (2012).
Baird, B., Smallwood, J., Mrazek, M. D., Kam, J. W. Y., Franklin, M. S., & Schooler, J. W. (2012). Inspired by Distraction: Mind Wandering Facilitates Creative Incubation. Psychological Science, 23(10), 1117-1122. https://doi.org/10.1177/0956797612446024
2. For convergent thinking, effects were inconsistent: mind wandering with awareness only aided low-difficulty repeated items, while mind wandering without awareness hindered novel moderate-difficulty items.
3. The study only measured frequency and awareness of mind wandering. The study ignores its content (e.g., task-related vs. unrelated thoughts). This omission restricts insight into how mind wandering influences creativity, a gap noted in prior work (Zedelius & Schooler, 2016).
Zedelius CM and Schooler JW (2016) The Richness of Inner Experience: Relating Styles of Daydreaming to Creative Processes. Front. Psychol. 6:2063. doi: 10.3389/fpsyg.2015.02063
4. Brain Sciences emphasizes neuroscience, yet this study is purely psychological, lacking neuroscientific methods or measures (e.g., EEG, fMRI). While the journal accepts psychological research, the absence of brain-related data makes it a poor fit.
5. The lack of correlation between divergent and convergent performance is intriguing but underexplored.
Reviewer 2 Report
Comments and Suggestions for Authors
I thank the authors for the privilege of reviewing “The role of mind wandering during incubation in divergent and convergent creative thinking”. This is a well written piece of work. The introduction is well written and provides a comprehensive and robust introduction of the current scope of research. I would like to commend the authors on an excellent piece of scholarship. The manuscript is well-written, thoughtfully structured. Additionally, the literature review is comprehensive and well-integrated, situating the study within the existing body of work while identifying important gaps that the current study addresses. However, what didn’t come out as prevalent is the existing gap which motivates this study. Perhaps the authors can place greater emphasis on this. The results, as written, supports previous studies. Overall, the manuscript demonstrates a high level of academic quality and originality. I have no major concerns or revisions to suggest at this time, and I recommend it for publication without hesitation.
Reviewer 3 Report
Comments and Suggestions for Authors
The authors present an interesting study testing the effects of mind wandering on divergent and convergent creative thinking. To this end, experimental manipulations are employed and, further, correlational analyses conducted. The sample size is adequate for both, in particular for the experimental design. The analyses are straightforward (conventional) and generally sound. The findings are informative, although results are somewhat mixed, suggesting an effect of conscious mind-wandering on convergent thinking only in some conditions. Overall the manuscript reads very well, it is clearly structured and the tone is clear. There is also some fit with the scope of this journal as the paradigms used in this study have been employed in behavioral neuroscience research. Below I would like to address some points that would need to be addressed in a revision.
Manipulation checks and possible moderation
- The choice of an 0-back task can be expected to leave sufficient resources for incubation (mind wandering). However, it would be more convincing to conduct a manipulation check.
- This should go beyond reporting an absence of mean differences for the two task types, but it should also test to what extent individual differences in the 0-back task possibly affect (i.e., moderate) the effects in the dependent variables.
- Further, correlations with self-reported mind wandering (thought probes) should be reported. (Not sure there are substantial relations. However, this would be not uncommon for performance and self-report measures.)
Choice of tasks and scoring
- I understand that the task have been selected because of their comparability in relevant features (verbal and similar duration). However, any task is only a specific indicator, suffering from unreliability and process-impurity. A conceptual replication with different tasks or a test of latent changes across a series of observed indicators would have been a possible remedy. While this may be beyond the scope of this contribution, it should be discussed as a limitation.
- The use of originality scores is delicate. If the score only denotes rare responses without a quality check, these may likely comprise both excellent and poor ideas. Consequently, an absence of relations with ability measures (as frequently found) would be expected. Possibly re-evaluate the responses in this light or, at least, discuss this as a limitation (obviously not only of the present contribution).
Power calculation and sample
- As far as I understand, the power calculations were conducted for main effects (either t-tests or repeated ANOVA). However, some of the effects (e.g., group differences [between] in change scores [within]) are conceptually interaction terms. The latter typically suffer from power and larger samples size would have been required.
- As least, discuss that the absence of differences between certain conditions (e.g., p=.12) rather reflect insufficient statistical power. I appreciate, though, that effect sizes are reported so the reader gets an impression of the magnitude of potential effects (irrespective of statistical significance).
- The sample is described as "broad" (l. 246). In fact, it comprises two clusters, students and a employees of food production enterprise. This will likely result in bimodal distributions of performance indicators, potentially resulting in inflates standard errors and biased correlations. Possibly double check with robust estimators that the overall pattern is comparable.
Avoid overinterpretation
- The idea to use test effects split by difficulty level sounds appealing, in particular if there are clear predictions. However, the very few items per condition (i.e., 2, 3, and 3) hardly suffice to derive reliable estimates. It does not come much as a surprise that the pattern appears difficult to reconcile with predictions.
- I suggest to report the interaction effect with item difficulty and the subsequent series of tests between specific conditions very cautiously. While this may change parts of the story, I believe it makes more sense to stay with the robust and (most likely) replicable results. Maybe, there is not that much of an effect of mind wandering at all, neither in divergent nor in convergent thinking?
Minor issues
- L. 240, l. 329: "related t-tests": maybe repeated t-test is more conventional?
- L. 362 and other places: "There were no effects of ...": Please report statistics also in case of a statistically non-significant effect.
- L. 388: "r=-0.47". I think leading zeros can be removed from correlation coefficients.
- l. 576 f: "participants must presumably have engaged in different types of reflection at different moments within it": Not sure what really contributed to the observed differences. I would suggest not to over-interpret.
- L. 613 ff (Limitations Section): In line with the points mentioned above, some more limitations should be (as the least) mentioned, in arbitrary order: (1) few items per difficulty condition, likely resulting in unreliable estimate, (2) likely insufficient power for interaction effects, (3) single task indicators, making it difficult to generalize, (4) possibly individual differences in effort completing the 0-back tasks, which may moderate the other relations, (5) clustered samples with possible effects on SE and relationship estimates. Naturally, some of these points could be checked empirically and rebutted.
Round 2
Reviewer 1 Report
Comments and Suggestions for Authors
The authors have not addressed my concerns appropriately.
Author Response
Comments 1: The authors have not addressed my concerns appropriately.
Response: In the absence of any detail as to what the reviewer feels has not been addressed appropriately, it is not possible for us to respond further.
Reviewer 3 Report
Comments and Suggestions for Authors
The authors have done an excellent job and adequately addressed most of the points and suggestions made in my previous review.
There is, however, one last issue that should be double-checked (l. 450 ff.): If response times and mind wandering were positively correlated, this would indicate that participants who responded more slowly engaged in more mind wandering (not the ones who responded more rapidly, i.e. with faster RTs). I think this relation is actually more plausible than the one currently mentioned in the text.
Apart from this, this manuscript is ready for publication in my eyes – and can be seen as an informative contribution to the literature on mind wandering.
Author Response
Comments 1: There is, however, one last issue that should be double-checked (l. 450 ff.): If response times and mind wandering were positively correlated, this would indicate that participants who responded more slowly engaged in more mind wandering (not the ones who responded more rapidly, i.e. with faster RTs). I think this relation is actually more plausible than the one currently mentioned in the text.
Our apologies - the word 'rapidly' was a somewhat odd typographical mistake, which we spotted very shortly after submission of the revised manuscript and tried - too late - to correct before the paper went back to reviewers. In the updated version, this mistake has been corrected, with the word 'rapidly' being replaced by 'slowly'. We thank the reviewer for their careful reading of the text!